



# Sensitivity and identifiability of rheological parameters in debris flow modeling

Gerardo Zegers[1,2], Pablo A. Mendoza[1,3], Alex Garces[1], and Santiago Montserrat[1]

[1]Advanced Mining Technology Center, Universidad de Chile
[2]Department of Geosciences, University of Calgary
[3]Department of Civil Engineering, Universidad de Chile

**Correspondence:** Santiago Montserrat (santiago.montserrat@amtc.cl)

**Abstract.**

Over the past decades, several numerical models have been developed to understand, simulate and predict debris flow events. Typically, these models simplify the complex interactions between water and solids using a single-phase approach and different rheological models to represent flow resistance. In this study, we perform a sensitivity analysis on the parameters of a debris

flow numerical model (FLO-2D) for a suite of relevant variables (i.e., maximum flood area, maximum flow velocity, maximum flow velocity, deposit volume). Our aims are to (i) examine the degree of model overparameterization, and (ii) assess the effectiveness of observational constraints to improve parameter identifiability. We use the Distributed Evaluation of Local Sensitivity Analysis (DELSA) method, which is a hybrid local-global technique. Specifically, we analyze two creeks in northern Chile that were affected by debris flows on March 25, 2015. Our results show that SD and $\beta_1$ - a parameter related to viscosity -

provide the largest sensitivities. Further, our results demonstrate that equifinality is present in FLO-2D, and that final deposited volume and maximum flood area contain considerable information to identify model parameters.

## 1 Introduction

In steep mountain environments, intense and localized storms can trigger the sudden movement of sediments, generating flash floods with solid volumetric concentrations up to 40 - 60 % (Takahashi, 1981; O'Brien, Jim S and Julien, 1988; Calvo and

Savi, 2009). These events, also known as debris flows, differ from water floods because - in addition to fluid stress - solid-fluid and solid-solid interactions dominate the flow motion (Takahashi, 1981; Iverson et al., 1997). In recent years, debris flows have been recognized as a major natural hazard (Calvo and Savi, 2009), affecting infrastructure, economic activities and human life. For instance, debris flows events in Switzerland produced 24 fatalities and overall losses of US $ 380 MM during the period 1972 - 2007 (Hilker et al., 2009). In Chile, estimated economic losses associated to the five biggest debris flow events recorded

over 1980-2017, were at least US $ 1.600 MM, with nearly 1,000 people dead or missing (Servicio Nacional de Geología y Minería, 2017).

Over the last decades, numerical models have emerged as a powerful tool to understand the behaviour and magnitude of debris flow events, since they allow the quantification of key variables used by engineers and decision-makers for risk management (Quan Luna et al., 2011; Frey et al., 2016; Calvo and Savi, 2009) and urban planning (Hürlimann et al., 2006;





Lucà et al., 2014; Naef et al., 2006; Arattano et al., 2006). However, the application of debris flow models requires several assumptions and simplifications that make results diverge from reality at various levels (Sosio et al., 2007). For example, uncertainties in terrain elevation models (e.g., satellite product, horizontal resolution), physical parameters (e.g., rheology parameters, solid concentration, and hydrological fluxes (e.g., precipitation, streamflow) used to force debris flow simulations can substantially impact relevant variables, such as flood area, sediment volumes, or maximum flow depth.

The use of debris flow models for practical problems typically requires the implementation of single-phase numerical models (Rickenmann et al., 2006; Naef et al., 2006) that solve 1D or 2D Saint Venant equations, using different rheological approaches to account for frictional stress ($S_f$). Many studies have reported good agreement between debris-flow model results and post-event measurements - e.g., run-out distance, flow velocity, deposit depth, flood area (D'Agostino and Tecca, 2006; Naef et al., 2006; Sosio et al., 2007; Rickenmann et al., 2006; Cesca and D'Agostino, 2008; Lin et al., 2011; Hungr, 1995). Nevertheless,

it is recognized that, because complex debris flow dynamics change in time and space (Coussot and Meunier, 1996), the appropriate choice of rheological parameters is critical for a good agreement between debris flow model output and field data (Sosio et al., 2007). In this context, various approaches have been adopted to characterize the sensitivity of debris flow model results to variations in model parameters - i.e., the coefficients in the model equations. For example, D'Agostino and Tecca (2006) compared FLO-2D (O'Brien and Garcia, 2009) simulations performed with two sets of rheological parameters

and three values of the laminar coefficient $K$ (6 simulations in total), concluding that $K$ controls the flood area and that rheological parameters control the maximum depth. Boniello et al. (2010) compared FLO-2D model results from a set of 12 back-calculated rheological parameters selected from previous studies, with another set of parameter values obtained from laboratory rheological analyses, finding a better representation of debris flow behavior with back-calculated parameters. Chow et al. (2018) conducted simulations with FLO 2D using 26 different sets of rheological parameters obtained from previous

studies, combined with different values of volumetric sediment concentration $C_v$, specific gravity $G_s$ and the surface detention $SD$, a parameter used in FLO2D to represent flow detention. They found that the most important parameters were $C_v$, $SD$ and $\beta_1$, which characterizes fluid viscosity. All these studies used fixed sets of rheological parameters in their numerical experiments and, therefore, the relative importance of such coefficients on relevant simulated variables - specifically, flow depth, flow velocity, deposit volume, and flood area - remains unkown. Therefore, this paper addresses the following questions:

1. How sensitive are debris flow model results to uncertain - and typically fixed - rheological parameters?

   2. What are the most effective post-event measurements to constrain the parameter search towards more realistic simulations?

   To answer these questions, we perform a sensitivity analysis on the parameters of a numerical debris flow model, and examine the effects of using post-event in-situ measurements on expected parameter ranges. In particular, we analyze a debris

flow event occurred in two creeks located in the Atacama region (northern Chile) during March 2015. Such event was the consequence of a heavy precipitation event over a three-day period, which exceeded 60 mm at several locations (Bozkurt et al., 2016), producing loss of human lives and massive infrastructure damage. Therefore, our intention is to provide guidance on the choice of uncertain rheological parameters, contributing to more reliable numerical simulations for debris flow risk assessments





and land use planning. The remainder of this paper is organized as follows: Section 2 describes the case study creeks and data,
Section 3 describes the numerical debris flow model, sensitivity analysis and parameter search strategies; Section 4 presents
the results and discussion, and Section 5 summarizes our main findings.

## 2   Study domain and data

We choose two nearly located ephemeral creeks in the upper Huasco River basin, Acerillas and La Mesilla (Fig. 1 (a)), where
debris flows were triggered by an extreme precipitation event on March 24-26, 2015 (Bozkurt et al., 2016; Ortega et al., 2019).
The Huasco Valley is a semi-arid fluvial system located at the southern edge of the Atacama region, Chile. This valley is
characterized by perennial rivers that only exist in the trunk valleys, while tributaries only show ephemeral streams. In these
areas, heavy rainfall events may induce catastrophic debris flows and mud-floods that greatly contribute to erosion (Aguilar
et al., in review, 2019).

   The Acerillas creek (15 $km^2$, Figure 1 (b)) has a markedly narrow channel with almost none alluvial fan, allowing the
transportation of sediments towards the El Carmen River. Post-event measurements indicate a deposited sediment volume of
6.000 $m^3$ (Cabré et al., 2020) and a maximum flood area of 37.000 $m^2$. Conversely, the La Mesilla creek (2,5 $km^2$, Figure
1 (c)) is characterized by a big alluvial fan where considerable sedimentation occurs, and post-event measurements show a
deposited sediment volume of 102.000 $m^3$ (Cabré et al., 2020) and a maximum flood area of 246.500 $m^2$. These flood areas
were estimated by comparing pre- and post-event satellite Google Earth imagery. Also, a post-event topography lidar scan
(acquired in Feb-March 2017 by the Chilean Ministry of Public Works) is available for this study. This dataset has a $1x1\,m^2$
horizontal resolution, and was post-processed in order to eliminate vegetation and buildings.

   Flow discharge data at the outlet of each creek were obtained from a distributed hydrological model, HEC-HMS version 4.2
(USACE, 2015). The model was configured for the entire Huasco River basin (7242 $km^2$), upstream the Santa Juana irrigation
reservoir, as part of a debris flow mitigation project for the Chilean Ministry of Public Works. Hydrologic model simulations
were forced using data from point measurements at 14 meteorological stations, spatially distributed with the inverse distance
weighting (IDW) interpolation method (Teegavarapu et al., 2006). Total rainfall records range from 20 $mm$ to 76 $mm$, with
a maximum registered intensity of $16\,mm\,hr^{-1}$. The HEC-HMS model parameters were calibrated against hourly streamflow
observed at two gauge stations located in the upper part of the basin - *Río El Carmen en El Corral* and *Río Conay en Las Lozas*
(Fig. 2) - obtaining a Nash-Sutcliffe efficiency (NSE; Nash and Sutcliffe, 1970) of 0.78 for the former and 0.64 for the latter.
Although all the other gauging stations were buried or destroyed by debris flows, simulated total water volumes were similar
with those captured by the Santa Juana Reservoir - whose levels were low before the event. Estimated peak flow discharges for
the event analyzed are $8\,m^3\,s^{-1}$ at La Mesilla and $12.7\,m^3\,s^{-1}$ at Acerillas.


**Figure 1.** (a) Location of the two case study creeks and reference models results. The maximum observed flood areas and modeled flow depth (reference models) are shown for (b) Acerillas creek, and (c) La Mesilla creek. Elevations bands created from Satellite DEM 12 x 12 m: © JAXA/METI ALOS PALSAR L1.0 2007. Accessed through ASF DAAC 11 June 2017.

## 3    Methods

### 3.1    Debris flow model

We use the two-dimensional FLO-2D debris flow model (O'Brien, James S and Julien, Pierre Y and Fullerton, 1993), configured at a 10-m horizontal resolution. FLO-2D is a finite difference model that simulates water or debris flows in channels or unconfined surfaces. The governing equations solved by FLO-2D are the depth-averaged continuity and momentum conservation (Eqs. (1) and (2)), and the flood wave progression is controlled by topography and flow resistance (O'Brien, Jim





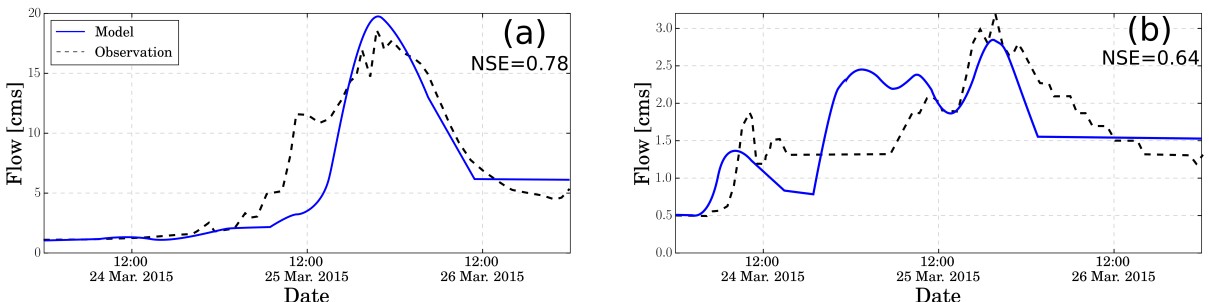

**Figure 2.** Calibration records for stream gauge stations (a) *Río El Carmen en El Corral* and (b) *Río Conay en Las Lozas*.

S and Julien, 1988). FLO-2D can also simulate debris-flows rheologies using a "quadratic" rheological model that combines

components associated with creep, viscous, dispersive (collisions), and turbulent stresses (O'Brien, Jim S and Julien, 1988; O'Brien and Garcia, 2009; Naef et al., 2006). Based on this quadratic rheology, the friction slope, $S_f$, is estimated as (Eq. (3)):

$$\frac{\partial h}{\partial t} + \frac{\partial hV_x}{\partial x} + \frac{\partial hV_y}{\partial y} = 0 \tag{1}$$

$$
\begin{aligned}
S_{fx} &= S_{ox} - \frac{\partial h}{\partial x} - \frac{V_x}{g}\frac{\partial V_x}{\partial x} - \frac{V_y}{g}\frac{\partial V_x}{\partial y} - \frac{1}{g}\frac{\partial V_x}{\partial t} \\
S_{fy} &= S_{oy} - \frac{\partial h}{\partial y} - \frac{V_y}{g}\frac{\partial V_y}{\partial y} - \frac{V_x}{g}\frac{\partial V_y}{\partial x} - \frac{1}{g}\frac{\partial V_y}{\partial t}
\end{aligned}
\tag{2}
$$

$$S_f = \frac{\tau_y}{\gamma_m h} + \frac{K\eta V}{8\gamma_m h^2} + \frac{n_{td}^2 V^2}{h^{3/4}} \tag{3}$$

where $h$ is the local flow depth, $t$ is time, $V_x$ and $V_y$ are depth-averaged velocity components along the $x$ and $y$ directions, $g$ is gravitational acceleration, $S_f$ the friction slope, $S_o$ is the bed slope, $\tau_y$ is the yield stress, $K$ a laminar strength parameter, $\eta$ the interstitial fluid dynamic viscosity, and $n_{td}$ is the conventional Manning's roughness coefficient corrected by $C_v$ ($n_{td} = 0.0538 n e^{6.0896 C_v}$ (O'Brien, Jim S and Julien, 1988)). O'Brien, James S and Julien, Pierre Y and Fullerton (1993) proposed the following empirical relationships to calculate the viscosity and yield stress as a function of the volumetric sediment

concentration, $c_v$ (Eqs. (4) and (5))

$$\eta = \alpha_1 e^{\beta_1 c_v} \tag{4}$$

$$\tau_y = \alpha_2 e^{\beta_2 c_v} \tag{5}$$

where $\alpha_{1,2}$ and $\beta_{1,2}$ are experimentally defined empirical coefficients (O'Brien, James S and Julien, Pierre Y and Fullerton, 1993; O'Brien and Garcia, 2009).



## 3.2 Sensitivity analysis

Sensitivity analysis (SA) is a powerful tool to characterize the effects of variations in input factors on environmental model responses (Razavi and Gupta, 2015; Gupta and Razavi, 2018). When the factors of interest are the model parameters, SA helps to identify those that are redundant for the modeling purposes, contributing to a more efficient parameter search (Mendoza et al., 2015). Different types of SA techniques have been proposed in the literature depending on specific objectives, or even the meaning of sensitivity (see, for example, reviews by Razavi and Gupta (2015); Pianosi et al. (2016)).

In this work, we apply the DELSA method (Rakovec et al., 2014) to identify the parameters that have the largest impact on simulated debris flow variables. DELSA is a frugal local-global hybrid technique, that provides first-order sensitivity indices across the parameter space. These measures are obtained using local gradients that quantify the sensitivity of a modeled output, $\Psi$, relative to individual variations of a parameter $\theta_j$. The local gradients, $\frac{\partial \Psi}{\partial \theta_j}|_k$, are used to compute the first-order sensitivity of each parameter $j$ at each point $k$ of the parameter space:

$$S1_k^j = \frac{|\frac{\partial \Psi_{kj}}{\partial \theta_j}|^2 \frac{1}{12}(\theta_{i,max} - \theta_{i,min})^2}{V_K(\Psi)} \tag{6}$$

where $V_K(\Psi)$ is the total local variance at point $k$:

$$V_K(\Psi) = \sum_{j=0}^{J} |\frac{\partial \Psi_{kj}}{\partial \theta_j}|^2 \frac{1}{12}(\theta_{i,max} - \theta_{i,min})^2 \tag{7}$$

The first-order sensitivity measures, $S1_k^j$, vary between 0 and 1, and the sum of first-order sensitivities from all parameters is equal to 1 at each sample point. In this work, parameter sampling is performed using the Latin hypercube sampling (LHS) method. LHS is a statistical method to generate an almost random sample of parameter values from a multidimensional distribution, and has proven to be more efficient than other methods like Monte Carlo sampling (Olsson et al., 2003; Olsson and Sandberg, 2002).

Local sensitivities can be analyzed in a disaggregated manner - through their cumulative frequency distribution across the parameter space -, or aggregated by computing a specific statistical property - e.g., the median of all local sensitivity measures for a particular pair of parameter and target variable. We use both approaches to analyze SA results.

In this paper, we focus on the effects of debris flow model parameters on four response variables: maximum average runoff speed $V_{mean}\, m\, s^{-1}$, maximum average runoff height $H_{mean}\, m$, maximum flood area $A_{max}\, m^2$, and deposited volume $Vol_{dep}\, m^3$. These response variables are calculated using the outputs from FLO-2D as:

$$V_{mean} = \sum_{j=0}^{N_{WC}} V_{max}(j)/N_{WC} \tag{8}$$


$$H_{mean} = \sum_{j=0}^{N_{WC}} H_{max}(j)/N_{WC} \tag{9}$$

$$A_{max} = \sum_{j=0}^{N_{WC}} dx * dy \tag{10}$$

$$Vol_{dep} = \sum_{j=0}^{N_{WC}} H_{final}(j)dx * dy \tag{11}$$

where $j$ is the cell index, $N_{WC}$ is the total number of wet cells ($h > 0$), $dx$ and $dy$ indicate the cell size along the x and y axis (numerical grid); $V_{max}(j)$, $H_{max}(j)$ and $H_{final}(j)$ are the maximum flow speed, maximum flow depth and final runoff height of the cell $j$ respectively

The FLO-2D parameters considered for DELSA are those that describe the fluid rheology (Table 1): $\alpha_{1,2}$, $\beta_{1,2}$, $C_v$, $K$, $n$, the $SD$ parameter and the total volume of sediments mobilized $V_T$.

The detention coefficient, $SD$, is a model parameter that reproduces flow detention. However, there is little information in the literature on how this parameter operates. Past studies D'Agostino and Tecca (2006) suggest that $SD$ acts as a minimum depth for flow routing (or non - flow condition). However, modeled flow depth can be lower than $SD$, suggesting that a second restriction exists for flow detention, probably related with flow velocity. D'Agostino and Tecca (2006) noted that this coefficient has a strong influence on the results and it can be used as a surrogate of the rheology. The total sediment volume mobilized ($V_{Test}$) by each debris flow event was estimated with the equation proposed by Chang et al. (2011) (Eq. (12)):

$$V_{Test} = 0.023A_w + 0.064A_L + 13264.6GI - 1399.2D + 38.47C_R \tag{12}$$

where $A_w$ is the watershed area, $A_L$ the landslide area (zero in these cases), $GI$ the geological index (where a value of 2.5 is assumed based on a study zone report made for the Chilean Ministry of Public Works), $D$ rainfall duration ($D = 48\,hours$ for this event) and $C_R$ the cumulative rainfall ($C_R = 76\,mm$). We obtain $V_{Test}$ values of $185000\,m^3$ for Acerillas and $154000\,m^3$ for La Mesilla.

Debris flow concentration is assumed to vary with streamflow between a minimum concentration $C_{vmin} = 0.1 - 0.4$ to a maximum concentration $C_{vmax}$ at the time of peak flow, which is treated as a model parameter. To this end, we propose the following function for $C_v$:

$$C_v(t) = \begin{cases} \frac{(C_{vmax}-C_{vmin})\cdot erf((Q(t)-Q_m)/(Q_{max}-Q_m)\cdot\phi)\cdot(C_{vmax}-C_{vmin})}{((C_{vmax}-C_{vmin})\cdot erf((Q_{max}-Q_m)/(Q_{max}-Q_m)\cdot\phi))} + C_{vmin}, & \text{if } Q(t)/Q_{mean} \geq 0.5 \\ C_{vmin}, & \text{if } Q(t)/Q_{mean} < 0.5 \end{cases} \tag{13}$$





**Table 1.** Values range of the model parameters.

| Parameter | MIN | MAX | Units | Reference |
|:---:|:---:|:---:|:---:|:---:|
| $\alpha_1$ | 0,00030 | 0,06480 | $poises$ | Sosio et al. (2007); O'Brien, Jim S and Julien (1988) |
| $\beta_1$ | 6,20 | 33,10 | - | O'Brien, Jim S and Julien (1988) |
| $\alpha_2$ | 0,00071 | 0,15200 | $dynes\,cm^{-2}$ | O'Brien, Jim S and Julien (1988); D'Agostino and Tecca (2006) |
| $\beta_2$ | 16,90 | 29,80 | - | O'Brien, Jim S and Julien (1988) |
| $Cv_{max}$ | 0,45 | 0,60 | - | Sosio et al. (2007); O'Brien, Jim S and Julien (1988) |
| $n$ | 0,01 | 0,2 | - | Rickenmann et al. (2006) |
| $K$ | 24 | 2000 | - | O'Brien and Garcia (2009) |
| $SD$ | 0,1 | 1,5 | - | O'Brien and Garcia (2009); D'Agostino and Tecca (2006) |
| $V_T$ | 70% $V_{Test}$ | 130% $V_{Test}$ | $m^3$ | Chang et al. (2011) |
| $\tau_y$ | 153,6 | 35.000,0 | $dynes\,cm^{-2}$ | O'Brien, Jim S and Julien (1988); Rickenmann et al. (2006) |
| $\eta$ | 1,1 | 100.000,0 | $poises$ | O'Brien, Jim S and Julien (1988); Rickenmann et al. (2006) |

where $\phi$ is a coefficient that changes the shape of the concentration curve. $\phi$ and $C_{vmin}$ are calculated in order to match the

total volume and minimize $C_{vmin}$ value:

$$\begin{aligned} \text{minimize} \quad & C_{vmin} \\ \text{subject to} \quad & \int_{t=0}^{T} C_v(t) \cdot Q(t) \cdot dt = V_T \end{aligned} \qquad (14)$$

Rheological parameter ranges are obtained from previous debris-flow studies (O'Brien, Jim S and Julien, 1988; Boniello et al., 2010; D'Agostino and Tecca, 2006; Sosio et al., 2007; Rickenmann et al., 2006; Chang et al., 2011; O'Brien and Garcia, 2009) and are summarized in Table 1. However, additional restrictions are imposed for $\tau_y$ and $\eta$, with maximum values of

35,000 $dynes\,cm^{-2}$ and 100,000 $poises$, respectively (Rickenmann et al., 2006). Since $\tau_y$ and $\eta$ are function of rheological parameters (Eqs (5) and (4)), such limits impose restrictions for $\alpha_{1,2}$, $\beta_{1,2}$ and $C_{vmax}$ that we ensure to implement in DELSA. To this end, we develop a Python script that allows running FLO-2D in parallel and sequentially, reducing computational cost considerably. For $V_T$, we assume a range of variation of $\pm 30\%$ respect to estimated values (Eq. (12)).

### 3.3 Parameter Selection via Constrained Search

We explore the effects of parameter uncertainty on simulated debris flow variables at the two case study creeks. We also examine the utility of using reference values for specific variables to constrain the search of physically plausible parameter sets. Such values are obtained from a reference simulation conducted by Zegers (2017), who validated a FLO-2D debris flow model at Acerillas and La Mesilla creeks using data from the 2015 flood event. Zegers (2017) calibrated model parameters by contrasting results against measured flood areas, deposited volumes and flow velocity estimated from a video captured with

a cell phone by a local person at Acerillas. Such validation strategy has provided reliable results for several other creeks in





**Table 2.** Parameter values for reference models on (Zegers, 2017).

| $\alpha_1$ | $\beta_1$ | $\alpha_2$ | $\beta_2$ | $Cv_{max}$ | $n$ | $K$ | $SD$ | $V_T$ |
|---|---|---|---|---|---|---|---|---|
| *poises* | — | *dynes cm$^{-2}$* | — | — | — | — | — | $m^3$ |
| 0,0075 | 14,39 | 0,152 | 18,7 | 0,55 | 0,07 flood plains<br>0,05 main channel | 2600 | 1 | 185.000 for Acerillas<br>154.000 for La Mesilla |

the area. Parameter values used for the reference simulation are provided in Table 2. Modeled flood areas and maximum flow depth are shown in Fig. 1, and discrepancies with respect to observations can be attributed to the use of post-event topography.

Based on the reference simulation, we choose reference values of $V_{mean} = 1\,m\,s^{-1}$ and $H_{mean} = 1.5\,m$ at both creeks. Additionally, we estimate the reference maximum flood area using Google Earth Imagery - obtaining values of 246500 $[m^2]$
for La Mesilla, and 37000 $[m^2]$ for Acerillas -, and use deposited sediment volumes reported by Cabré et al. (2020) as reference values, which correspond to 102000 $[m^3]$ and 6000 $[m^3]$ for La Mesilla and Acerillas, respectively.

## 4 Results and discussion

### 4.1 Choice of sample size

First, we seek to identify the minimum sample size, $N_k$, for which stable DELSA results can be obtained in order to minimize
computational cost (Rakovec et al., 2014). Therefore, we explore the effects of the choice of $N_k$ on the cumulative frequency distributions (CDFs) of DELSA first order sensitivity indices. Since we include $N_j = 9$ parameters, the total number of simulations required for each case is $N_t = (N_j + 1)N_k$. Figure 3 illustrates the sensitivity of DELSA results to variations in sample size, $N_k$, for four variables simulated by FLO-2D, with respect to parameters $\beta_1$ and $C_{vmax}$, at the Acerillas creek. Since the curves obtained for $N_k = 500$ and $N_k = 1000$ show slight differences, departing from the CDF for $N_k = 100$, we conclude that
a sample size of $N_k = 500$ is adequate for further analyses. The sensitivity of DELSA results to $N_k$ was also examined at La Mesilla creek, obtaining the same conclusion regarding sample size (not shown).

Figure 3 also shows that, depending on the target variable and parameter analyzed, first-order sensitivity indices can be highly heterogeneous across the parameter space. In particular, the modeled response is highly sensitive to variations of $\beta_1$, with first-order sensitivities larger than 0.2 for approximately 60% of cases, and sensitivities greater than 0.5 for 20% -40%
depending on the variable analyzed. On the other hand, the modeled variables are less sensitive to $C_{vmax}$ in most cases, with DELSA indices smaller than 0.1 for approximately 70% of the parameter sets.

### 4.2 Sensitivity of model responses to model parameters

Figure 4 displays the median of the full frequency distribution (obtained with $N_k = 500$) of local first order sensitivity indices for the two study domains: (i) Acerillas creek (ACE), and (ii) La Mesilla creek (MES). The uncertainty bands are obtained
by performing bootstrapping with replacement (resampled 1000 times). In general, $\beta_1$ provides the largest sensitivities for the simulates variables analyzed, which means that the fluid rheology, in particular the viscosity coefficient ($\eta = \eta(\beta_1)$), is a main


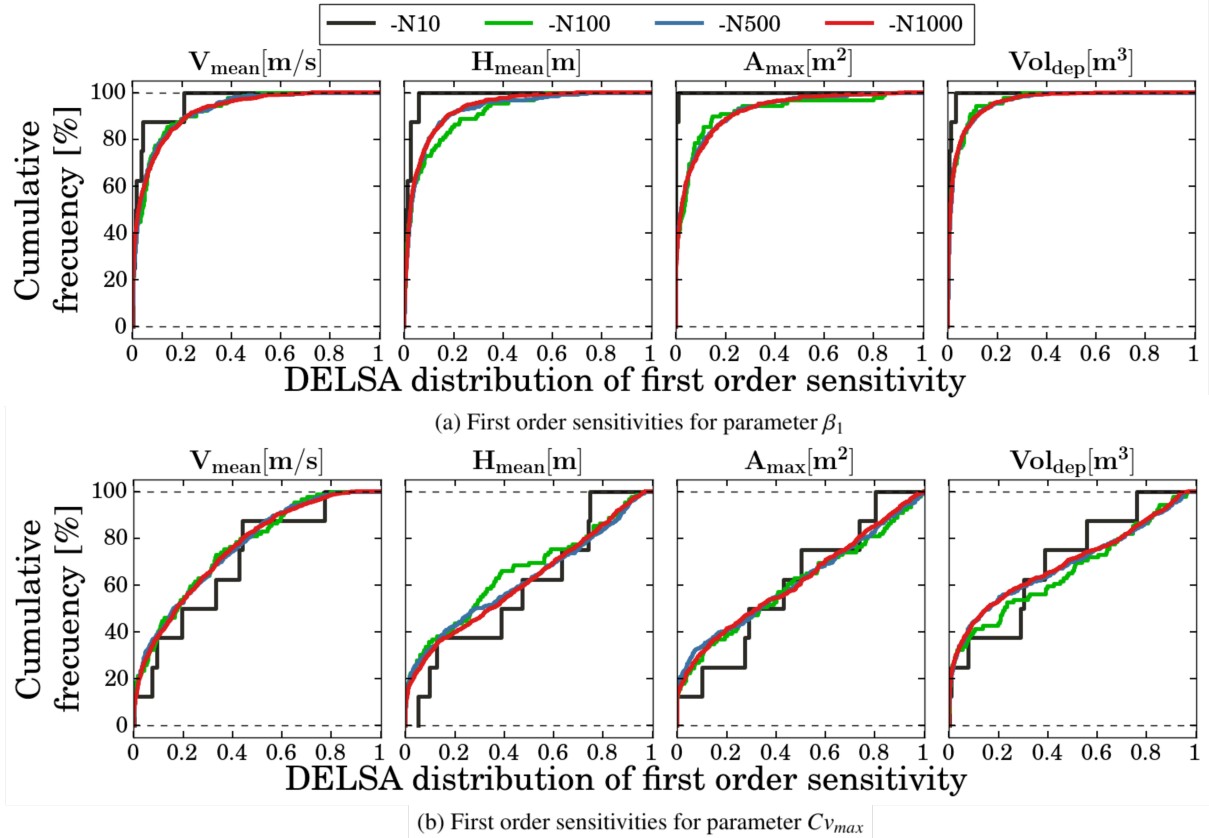

**Figure 3.** Effects of sample size $N_k$ on the cumulative distribution function (CDF) of DELSA indices for (top) parameter $\beta_1$, and (bottom) parameter $Cv_{max}$. These results illustrate the effects of parameter on flow velocity, flow depth, flood area, and deposited volume the Acerillas creek.

parameter controlling flow behaviour. Moreover, Acerillas' results show to be more sensitive than those obtained at La Mesilla with respect to $\beta_1$. This could be better explained when analysed together with $SD$, the detention coefficient.

As expected, the simulated deposited volume $Vol_{dep}$ is very sensitive to $SD$ because this parameter controls flow detention.
For the remaining simulated variables, $SD$ also rises as an important parameter at La Mesilla, but shows secondary importance at Acerillas, which can be explained by catchment differences. While the fluid rheology explains flow behavior at the Acerillas creek (mainly sensitive to $\beta_1$), depositional or detention processes - represented by $SD$ - gain importance across the larger alluvial fans of La Mesilla creek.

Another parameter that provides large sensitivities in simulated variables is the total sediment volume, $V_T$, especially in the
mean flow velocity, $V_{mean}$, whose sensitivity indices are of the same order of magnitude to those produced by $\beta_1$. However, $V_T$ does not produce large variations in the total deposited volume. As deposition occurs, the flow is channelized between the deposited margins, preventing flow spreading. For example, when increasing $SD$ values, $A_{max}$ decreases as flows are forced


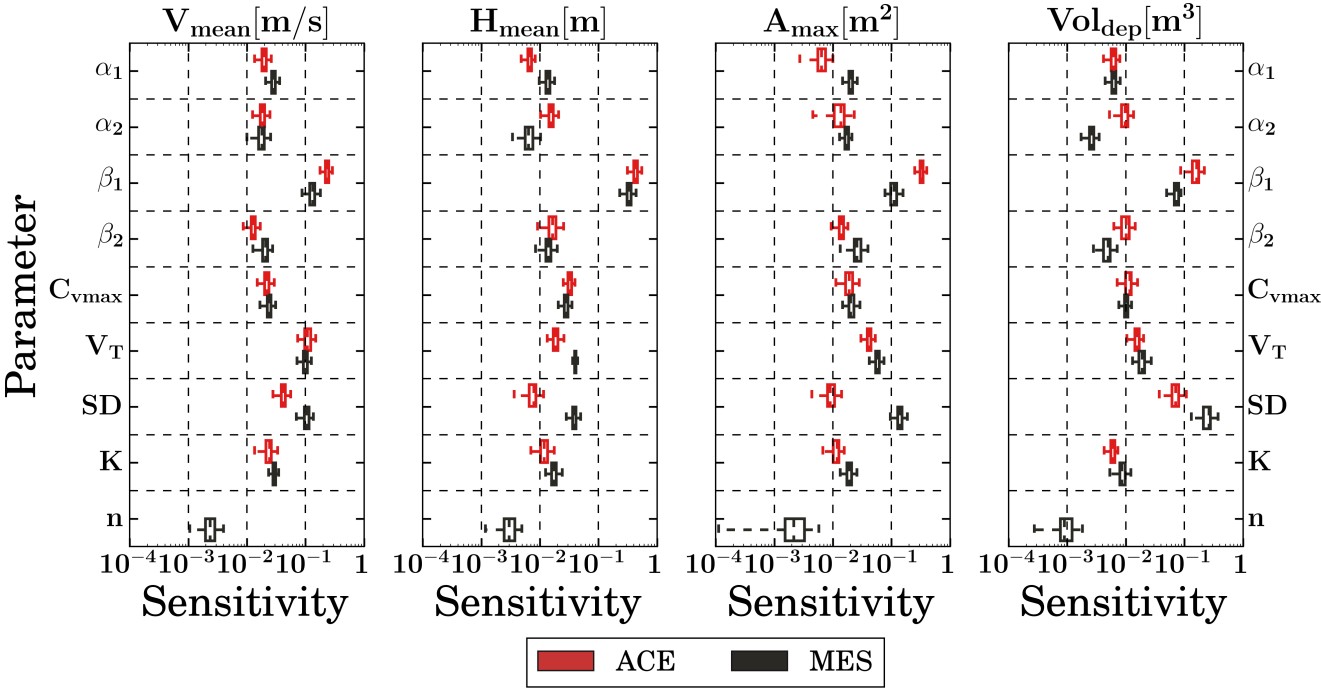

**Figure 4.** DELSA sensitivity indices (synthetized as the median from the cumulative frequency distribution) for all parameters and model responses. Results are displayed for Acerillas (red) and La Mesilla (black) creeks, and the sampling uncertainty (bootstrapping with 1000 times resampling) is indicated by boxplots. The vertical bold line in the boxplot is the median, the body of each boxplot shows the interquartile range (Q75 - Q25) and the whiskers represent the sample minima and sample maxima. DELSA indices are displayed in log space for a better visualization of inter-parameter differences.

to stop at deeper heights. However, this could be a structural weakness of FLO-2D, which lacks proper representations of complex depositional and dewatering processes.

Although sediment concentration is one of the main parameters controlling debris flow rheology, model results are insensitive to $C_{vmax}$. This is explained by its small range of variation compared to the feasible range of $\beta_1$, both parts of the exponential terms controlling fluid viscosity in Eq. (4). However, DELSA sensitivity indices associated with parameters determining yield stress ($\tau_y$) and Manning's roughness coefficient - both describing flow rheology -, are of second-order importance. In particular, model results are insensitive to Manning's roughness coefficient.

**4.3    Parameter uncertainty effects**

Figure 5 shows the full range of variation in model responses (produced by $N_k$ = 500 parameter sets), normalized by reference values obtained from post-event measurements ($Vol_{dep}$ and $A_{max}$) and results obtained from the reference models (for $H_{mean}$ and $V_{mean}$). These are compared with the simulated ensemble that results from screening model outputs imposing five different





observational constraints: (i) ±20% reference mean flow velocity (FVEL), (ii) ±20% reference mean flow depth (FH), (iii)
±20% reference maximum flood area (FAREA), (iv) ±40% reference volume deposit (FVOL) , and (v) ±20% reference
maximum flood area and ±40% reference volume deposit (FAREAVOL). To be clear, constraint (i) results from keeping all
those parameter sets that provide a simulated mean flow velocity within the range $0.8V_{ref} - 1.2V_{ref}$. Constraints (ii)-(iv) work
in a similar way for other observed variables, while constraint (v) filters all parameter sets that simultaneously provide flood
areas and deposit volumes within the ranges $0.8A_{ref} - 1.2A_{ref}$ and $0.6VOL_{ref} - 1.4VOL_{ref}$, respectively. We assume a
weaker observational constraint in the case of the deposited volume because of possible uncertainties associated to different
measurements techniques.

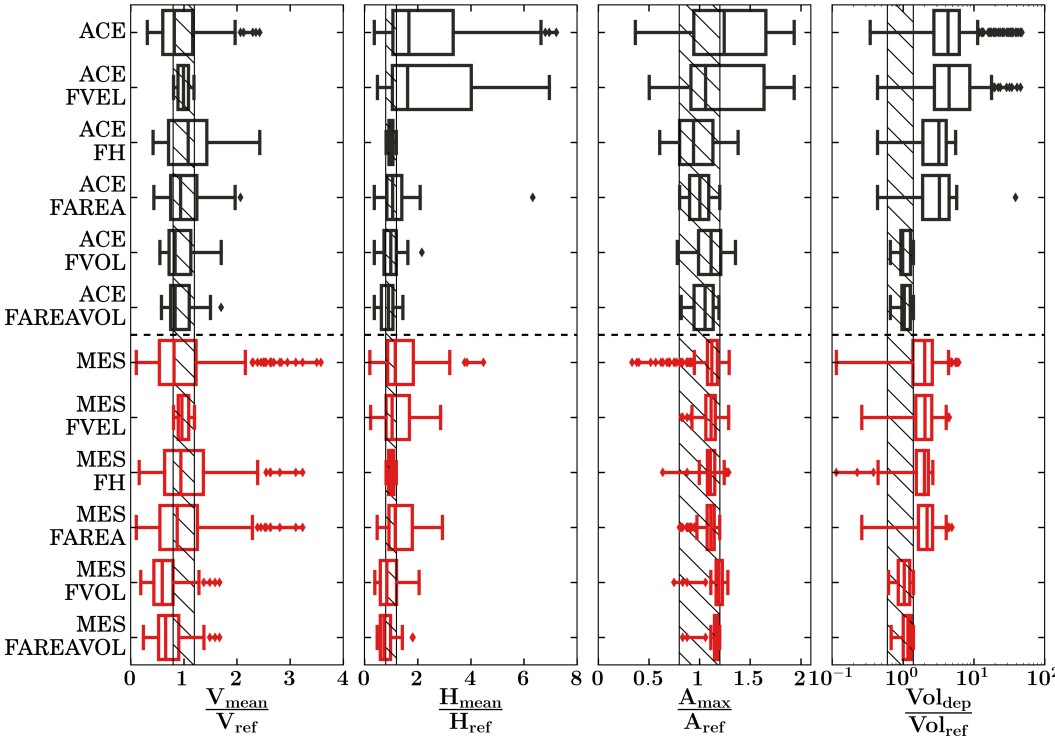

**Figure 5.** Effects of parametric uncertainty in normalized model responses for the original parameter sample (top panels) and alternative
observational constraints: flow velocity (FVEL), flow depth (FH), flood area (FAREA), and sediment volume (FVOL), and joint area-volume
constraint (FAREAVOL). Results are displayed for Acerillas (black) and La Mesilla (red) creeks. The hatched area for $V_{mean}$, $H_{mean}$ and
$A_{max}$ corresponds to ±20% of their reference values, and for $VOL_{dep}$ to ±40% of the estimated sediment volume. The vertical bold line
in the boxplots is the median, the body of each boxplot shows the interquartile range (Q75 - Q25) and the whiskers represent the sample
minima and sample maxima.

Figure 5 also shows that, in general, the effects of parametric uncertainty on simulated variables are considerable, and the
ensemble median can be substantially different from the reference boundaries. This is somewhat expected, since the literature





provides large ranges for model parameters (see Table 1 for details). Overall, uncertainties arising from the original parameter
samples (top panels) are larger in Acerillas, except for mean flow velocity. This could be explained by the larger sensitivity
of flow velocity to flow rheology, while the rest of simulated variables are more sensitive to deposition processes, mainly
represented by SD.

Most simulations overestimate deposited volumes and flow depth, especially at Acerillas. For flow velocity, the ensemble
of parameter sets provides mixed results in both creeks, with underestimation in most cases (median values lower than the
reference values); however, there are still several parameter sets that produce an overestimation of flow velocity. The results
obtained for maximum area reveal differences among both creeks: in Acerillas, most parameter sets tend to overestimate the
flood area, whereas most simulated values are within the expected range at La Mesilla. This could happen because $Vol_{dep} \ll$
$V_T$ at Acerillas, while $Vol_{dep} \sim V_T$ at La Mesilla; moreover, the maximum flood area at La Mesilla is approximately six
times larger than in Acerillas. Thus, small variations in $Vol_{dep}$ imply important fractional changes with respect to the volume
reference values at Acerillas.

Fig. 5 shows that the velocity constraint "FVEL" does not have an impact on the rest of simulated variables. Nevertheless,
the application of alternative observational constraints helps to reduce the spread of the remaining variables. For example, the
height filter "FH" improves simulations of maximum flood area, although it does not have much effect on velocity or deposit
volume. The area restriction, "FAREA", improves simulated flow depth only at Acerillas, as most of the original ensemble
members were already inside the expected reference boundaries at La Mesilla. The volume constraint "FVOL" reduces the
uncertainty in all variables, with the smallest improvement for flow velocity. This is because the volume is directly linked
to flow height and flood area, but not to flow velocity. Finally, the largest reductions in ensemble spread are obtained when
parameter sets are constrained by using area and volume observations (FAREAVOL).

The maximum flood area and deposited volume are relatively easy to measures, and are probably the most used post-event
measurements for calibrating debris flows models (Chow et al., 2018; Cesca and D'Agostino, 2008; D'Agostino and Tecca,
2006; Sosio et al., 2007; Frey et al., 2016; Quan Luna et al., 2011).

### 4.4   Parameter identifiability

Figure 6 illustrates the effects of applying observational constraints, specifically flood area and deposited volume constraints,
on parameter identifiability. Results show that the resampled values of $\alpha_{1,2}$, $\beta_2$, $C_{vmax}$, $V_T$, $K$ and $n$ cover practically the
entire original range. However, the application of observational restrictions provides substantial reductions in the ranges of
$\beta_1$ and $SD$, which mainly explain the flow rheology and depositional processes in our study areas. Further, lower parameter
values are obtained in comparison to the full range, especially $SD$ at the Acerillas creek. These results indicate that viscosity
$\eta = \eta(\alpha_1, \beta_1, C_{vmax})$ is the most restricted parameter when applying these constraints, discarding all medium-high values. On
the other hand, re-sampled values of $\tau_y = \tau_y(\alpha_2, \beta_2, C_{vmax})$ cover almost the entire original range.
The reference $SD$-value is close to the upper range in La Mesilla after applying constraint FAREVOL, and also much
larger than the resulting maximum values filtered at Acerillas. This result is somewhat expected, since, $SD$ does not provide
large model sensitivities in that domain. Similarly, the reference $\beta_1$ value is within the upper range of filtered values ($> Q75$).




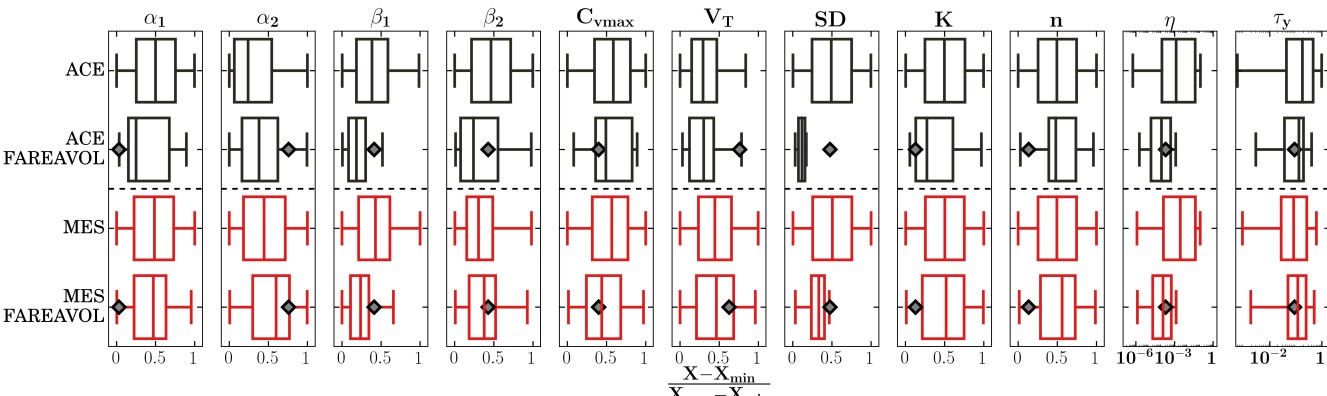

**Figure 6.** Effects of applying a flood area-volume observational constraint (FAREAVOL) on parameter identifiability. Results are displayed for Acerillas (black) and La Mesilla (red) creeks. The vertical bold line in the boxplots is the median, the body of each boxplot shows the interquartile range ($Q75 - Q25$) and the whiskers represent the sample minima and sample maxima. The grey-black diamonds represent parameter values for the reference simulations.

However, filtered $\eta$-values around the baseline model parameter result from the compensation of low values of $\alpha_1$ (near the minimum in the filtered range) and $C_{vmax}$ (below the median of filtered values). A similar effect is observed for $\tau_y$, whose

filtered range results from the compensation of $\alpha_2$, $\beta_2$ and $C_{vmax}$. In summary, different combinations of $\alpha_{1,2}$, $\beta_{1,2}$, and $C_v$ can generate viscosity and yield stress values that are suitable to reproduce the 2015 debris flow events in Acerillas and La Mesilla. This is a well-known problem in environmental models - referred to as equifinality, nonuniqueness or nonidentifiability of model parameters - that has been widely discussed for more than three decades in the hydrology literature (e.g., Beven 2006; Kelleher et al., 2017 ), but not carefully addressed in the debris flow modeling community.

Figure 6 also shows different behavior in other parameters. For example, the reference values for $K$ and $n$ are in the lower range of the filtered ensembles, while the reference $V_T$ is in the upper body of the boxplot. Low $K$-values produce low $S_{f2}$, the second term at the right hand of Eq. (3), representing viscous stress. This could be compensated with larger values of $SD$, as in the reference model. These results demonstrate that equifinality in FLO-2D does not only involve rheological parameters, and that $SD$ could be an important parameter to correct unrealistic model representation of rheology (D'Agostino and Tecca,

280 2006).

## 5   Summary and conclusions

We performed a sensitivity analysis on the parameters of a widely used numerical debris flow model (FLO-2D) and assess the effects of applying observational constraints on parameter identifiability. Our study domains are two morphologically different ravines, *Acerillas* and *La Mesilla*, located in the Atacama region, Chile. While Acerillas is characterised by a straight and well

defined channel with almost non alluvial fans, La Mesilla has a big alluvial fan where deposition is prone to occur.





We found that $\beta_1$, a parameter used to estimate the fluid mixture viscosity, provides the largest sensitivities in the variables analyzed followed by $SD$, a model parameter used to represent flow detention. Interestingly, the relative importance of $\beta_1$ and $SD$ depends on the study site, being the former more important for Acerillas and the latter for La Mesilla. These results suggest that, while rheological processes dominate flow behaviour at Acerillas (straight channel with small alluvial fan), sedimentation

and detention processes control flow in La Mesilla (big alluvial fan). Additionally, the total mobilized sediment $V_T$ is an important parameter for representing flow velocity, which is in line with previous evidence that higher flow volumes reduced effective friction coefficients in granular flows, increasing the overall fluid mobility (Johnson et al., 2016; Montserrat et al., 2016). Model results show to be almost insensitive to $n$, while DELSA sensitivities for the remaining parameters are of second order importance and provide similar indices.

The comparison between the original model parameter ranges (N = 500) and the ensemble resulting from applying observational restrictions shows that $SD$ and $\beta_1$ (i.e. $\eta$) are the parameters whose identifiability is mostly improved, while others practically preserve their original range. In addition, we obtain that different combinations of model parameters (including those that describe rheology) can provide very similar results, indicating that equifinality is present in FLO-2D. Our results also support the idea that single phase rheological models lack a strong physical basis (Iverson, 2003) and, therefore, their

determination requires expert knowledge. However, an encouraging finding is that the final deposited volume ($VOl_{dep}$) and maximum flood area ($A_{max}$) contain considerable information to identify model parameters.

We obtain that $SD$ strongly affects model results at La Mesilla, having also large effects on simulated deposited volumes at Acerillas. Moreover, this study provides evidence that $SD$ is one of the most important parameters controlling flow behaviour, and could possibly surrogate rheology in the model (D'Agostino and Tecca, 2006). One-phase debris flows models still lack

robust representations of complex process interactions during flow stopping that produce temporal and spatial changes in fluid rheology. Thus, these rheological changes have been replaced by simpler approaches (e.g., the incorporation of $SD$). Future work should advocate for improving debris flow models by incorporating better approaches to simulate deposition/erosion, stopping phases, and changing rheologies.

*Author contributions.* G.Z, A.G, P.M and S.M were involved on the conceptualization. G.Z, P.M and S.M contributed to the methodology,
analysis and drafted the manuscript. G.Z configured the model, conducted all simulations, analyzed the results and made the figures. G.Z, A.G, P.M and S.M contributed to writing - original draft. G.Z, P.M and S.M performed the writing - review & editing.

*Competing interests.* The authors declare that they have no conflict of interest.

*Acknowledgements.* The authors thank Tomás Gómez and Miguel Lagos for providing the streamflow data used for this study. We also thank the support of the Engineering Faculty of Universidad de Chile through the Advanced Mining Technology Center (AMTC), the Department



of Civil Engineering and IDIEM. The authors also thanks to the Public Work Ministry and the CONICYT/PIA Project AFB180004. PAM
acknowledges support from Fondecyt postdoctoral grant No. 3170079.



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
