# Peer review of "Sensitivity and identifiability of rheological parameters in debris flow modeling"

_Natural Hazards and Earth System Sciences, 2020_

## Referee Comment (RC1) · Martin Mergili (Referee) · 28 Mar 2020

Review to the paper

**Sensitivity and identifiability of rheological parameters in debris flow modeling**

Submitted to NHESS by *Gerardo Zegers* et al.

Reviewer: *Martin Mergili*

The authors investigate on the sensitivity of debris flow simulation results to the model parameters used by the FLO-2D software. Two debris flows in northern Chile are employed as case studies. Some of the key findings of the study are that there is some redundancy of information, that there is a certain degree of equifinality, hampering the identification of "correct" model parameter sets, and there is a broad spectrum of levels of sensitivity among the different model parameters with regard to the various reference parameters.

This paper covers a highly relevant topic which is certainly within the scope of NHESS. It is generally well-written, concise, and appropriately illustrated. Reference to the relevant sources is given. Before publication, I recommend some optimization, mainly concerning the precision of some formulations and statements and comparison of the results with other studies. In summary, I recommend **minor-moderate revisions**.

**General comment**

What would be interesting to see is a little bit more of discussion on how the findings of the study (e.g. the patterns shown in Fig. 4 and Fig. 5) relate to previous work. Do the results confirm earlier studies, or are there some contradictions? If yes, how could they be explained?

**Specific comments**

Section 3.1: some of the references are formatted in a strange way

L143: V is used here for volume, but before was is used for velocity - using $Vol_T$ instead of $V_T$ would be more consistent. The same applies to $V_{Test}$ introduced in L149.

L146: "or non-flow condition": please explain more clearly how this relates to SD.

L172ff: If Zegers (2017) has successfully simulated an event through calibrating the parameters with this same event, this is NOT a validation. It would only be a validation if the calibrated parameters are then applied to another event. Please clarify.

Fig. 1: Nice figure, but two remarks: (i) The lines in the overview pane leading to the detailed map of La Mesilla should pass behind the legend, and not in front of it. Further, it would look better if the lower line for La Mesilla would start at the southern end of the deposit. (ii) The threshold values in the legends are ambiguous: >0-1 m; >1-2 m etc. would be correct.

Fig. 3: Please revise caption (some issues of grammar).

This is all from my side. If the authors disagree with the one or the other comment, or would like to discuss issues, they should feel free to contact me at martin.mergili@univie.ac.at.

With best regards, Martin Mergili

---

## Author Comment (AC1) · 18 Apr 2020

We thank the Reviewer for his time in commenting on our paper. We provide responses to each individual point below. For clarity, comments are given in italics, and our responses are given in plain blue text.

The authors investigate the sensitivity of debris flow simulation results to the model parameters used by the FLO-2D software. Two debris flows in northern Chile are employed as case studies. Some of the key findings of the study are that there is some redundancy of information, that there is a certain degree of equifinality, hampering the identification of "correct" model parameter sets, and there is a broad spectrum of levels of sensitivity among the different model parameters with regard to the various

reference parameters. This paper covers a highly relevant topic which is certainly within the scope of NHESS. It is generally well-written, concise, and appropriately illustrated. Reference to the relevant sources is given. Before publication, I recommend some optimization, mainly concerning the precision of some formulations and statements and comparison of the results with other studies. In summary, I recommend minor moderate revisions.

***General comment*** What would be interesting to see is a little bit more of discussion on how the findings of the study (e.g. the patterns shown in Fig. 4 and Fig. 5) relate to previous work. Do the results confirm earlier studies, or are there some contradictions? If yes, how could they be explained?.

We will address this recommendation by adding more discussion on how our results and findings relate to previous work. Such discussion will be framed around the following ideas:

D'Agostino and Tecca (2006) compared FLO-2D simulations performed with two sets of rheological parameters and three values of the laminar coefficient K (six simulations in total), concluding that K controls the flood area and that rheological parameters control maximum flow depth. On the other hand, our results indicate that the laminar coefficient K provides the fourth-fifth largest sensitivities in simulated flood areas. Such difference probably relies on their experimental design, since their conclusions are drawn from only six model simulations.

Chow et al. (2018) conducted simulations with FLO 2D using 26 different sets of rheological parameters obtained from 45 previous studies, combined with different values of volumetric sediment concentration (Cv), specific gravity (Gs), and surface detention (SD). They found that the most influential parameters - in order of importance - were Cv, SD, and $\beta 1$ (which characterizes fluid viscosity). These results are different from ours due to discrepancies in the sampling method, their use of fixed sets of rheological parameters and their parameter ranking definition, which does not consider separate

effects on key simulated variables. Further, Chow et al. (2018) analyzed changes in Cv ignoring the effect of the total sediment volume, whereas we examine the effects of total volume and sediment concentration separately, obtaining that the total sediment volume is more important than the maximum sediment concentration. Finally, Chow et al. (2018) used fixed sets of parameters related to viscosity and shear stress, reporting thus an aggregate effect of a bigger variation of these parameters. In this study, we analyze parameter importance through local sensitivities across the entire parameter space, recognizing that the relative importance of parameters can change depending on the sub-region examined. This effect is shown in Fig. 1 through the cumulative frequency distribution of DELSA indices for Cvmax on the mean Velocity. For 60

**Specific comments**

*Section 3.1: some of the references are formatted in a strange way*

References will be formatted correctly, following the reviewer's suggestion.

L143: V is used here for volume, but before was is used for velocity - using VolT instead of VT would be more consistent. The same applies to VTest introduced in L149.

We will change the notation as suggested by the reviewer. we will use $Vol_T$ for the total volume and $Vol_{Test}$ for the estimated total volume of sediments.

L146: "or non-flow condition": please explain more clearly how this relates to SD.

We thank the reviewer for this observation. We agree that "or non-flow conditions" is confusing, so the text will be modified as follows: The detention coefficient SD is a model parameter that reproduces flow detention. The FLO-2D User's Manual and previous studies (D' Agostino and Tecca, 2006) suggest that $SD$ acts as the minimum flow depth possible to occur (i.e. flow stops if flow depth $< SD$). D'Agostino and Tecca (2006) noted that this coefficient has a strong influence on the results, and it can be used as a surrogate of the rheology.

L172ff: If Zegers (2017) has successfully simulated an event through calibrating the

parameters with this same event, this is NOT a validation. It would only be a validation if the calibrated parameters are then applied to another event. Please clarify.

We agree with the reviewer's point in that the word "validation" is misused. We will re-word the paragraph as follows: "Such values are obtained from a reference simulation conducted by Zegers (2017), who successfully simulated the 2015 debris flow events at Acerillas and La Mesilla creeks using FLO-2D.

Fig. 1: Nice figure, but two remarks: (i) The lines in the overview pane leading to the detailed map of La Mesilla should pass behind the legend, and not in front of it. Further, it would look better if the lower line for La Mesilla would start at the southern end of the deposit. (ii) The threshold values in the legends are ambiguous: >0-1 m; >1-2 m etc. would be correct.

We thank the reviewer's suggestions. The new Figure will look like:

Fig. 3: Please revise caption (some issues of grammar).

The figure caption will be corrected as suggested by the reviewer.

[Figure]

(a) First order sensitivities for parameter $\beta_1$

[Figure]

(b) First order sensitivities for parameter $Cv_{max}$

**Fig. 1.** Effect of the sample size Nk on the cumulative distribution function (CDF) of DELSA indices for parameters beta1 (top) , and Cv{max} (bottom) at the Acerillas creek.

**Fig. 2.** (a) Location of the two case study creeks and reference models results. The maximum observed flood areas and modeled flow depth (reference models) are shown for (b) Acerillas creek, and (c) La Mesilla

---

## Referee Comment (RC2) · Mary Hill (Referee) · 5 May 2020

Being able to match data with model results will always be an important part of people believing model results. However, those of us who understand what goes into developing a model and its application know that a good match to data is just the first step of how models can help people understand the many processes that define the world upon which our lives depend.

In this work, data and physical processes are modeled and explored. The results extend as far past just model fit as current technology supports. The DELSA method is used to reveal a considerable amount about which parameters are important to four defined metrics, and summarizes in some detail how this changes over parameter

space (fig. 3 and 4). The article also illustrates how new data can reduce parameter uncertainty and how this changes over likely parameter space (fig. 5). In both cases, the results provided by DELSA are a step towards being able to evaluate if the results suggest the model performs realistically – both because it fits the data reasonably well and because the parameters that are important and unimportant in different parts of parameter space make sense.

I have two points I would like to make in this review.

One is to highlight a potential of DELSA not noted in the paper. In line 117, this paper refers only to the first-order sensitivity capabilities of DELSA. While Rakovec et al (2014) first demonstrated the DELSA approach using first-order sensitivity indices, they also note that the approach has considerable unexplored potential for evaluation of parameter interactions. This requires that the sensitivity matrix include the prior information used for first-order statistics, and also derivatives related to observations, as noted in Rakovec et al (2014, paragraphs 11, 20, 66 and 67, and Figure 12 and Appendix A), and Hill and Tiedeman (2007, Appendix B).

The addition of observations in the sensitivity matrix allows calculation of statistics that address concerns such as those considered in Fig. 5 of this work. Commonly this is called a Value of Improved Information (VOII) analysis, and statistics such as OPR (Observation-PRediction) and PPR (Parameter-PRediction) could be used to explore the distribution of uncertainty measures throughout parameter space using the DELSA approach. OPR and PPR are described by Tiedeman et al. (2003, 2004), and Tonkin et al (2007). Parameter-value dependence of these or other statistics with similar goals has received little attention to my knowledge.

I imagine that the analyses in this article and those suggested are a small beginning of a future that will see models of complex processes used in ways we early modelers can scarcely imagine. We are riding on horseback while in the future there will be progressively more insightful ways to regard models and integrate their insights into

society. This is what I imagine. I am excited that I might live to see what will happen in a few coming decades.

My second comment is much shorter. In line 215 of the article we find the text "Although sediment concentration is one of the main parameters controlling debris flow rheology, model results are insensitive to Cvmax. This is explained by its small range of variation compared to the feasible range of beta-1, ..." I can see how a narrow parameter range can explain uniformity in parameter sensitivity, but fail to see how it can explain its insensitivity. Perhaps there is something not quite explained well here.

I appreciate the opportunity to provide comments on this very fine paper. I hope my comments provoke a bit and are of some utility.

References cited

Hill, M. C, Teideman, C .R. (2007) Effective groundwater calibration, with analysis of data, sensitivities, predictions, and uncertainty, Wiley-Interscience, John Wiley and Sons, New York, 455p.

Tiedeman, C. R., M. C. Hill, F. A. D'Agnese, and C. C. Faunt (2003), Methods for using groundwater model predictions to guide hydrogeologic data collection, with application to the Death Valley regional groundwater flow system,Water Resources Research, 39(1), 1010, doi:10.1029/2001WR001255.

Tiedeman, C. R., D. M. Ely, M. C. Hill, and G. M. O'Brien (2004), A method for evaluating the importance of system state observations to model predictions, with application to the Death Valley regional groundwater flow system, Water Resources Research, 40, W12411, doi:10.1029/2004WR003313.

Tonkin, M. J., C. R. Tiedeman, D. M. Ely, and M. C. Hill (2007), OPRPPR, A computer program for assessing data importance to model predictions using linear statistics, U.S. Geol. Surv. Tech. and Methods Rep. TM-6E2, 115 pp.

---

## Author Comment (AC2) · 20 May 2020

We thank Dr. Hill for her time in reviewing our paper. We provide responses to each individual point below. For clarity, comments are given in italics, and our responses are given in plain blue text.

*Being able to match data with model results will always be an important part of people believing model results. However, those of us who understand what goes into developing a model and its application know that a good match to data is just the first step of how models can help people understand the many processes that define the world upon which our lives depend. In this work, data and physical processes are modeled and explored. The results extend as far past just model fit as current technology*

[Figure]

*supports. The DELSA method is used to reveal a considerable amount about which parameters are important to four defined metrics, and summarizes in some detail how this changes over parameter paper space (fig. 3 and 4). The article also illustrates how new data can reduce parameter uncertainty and how this changes over likely parameter space (fig. 5). In both cases, the results provided by DELSA are a step towards being able to evaluate if the results suggest the model performs realistically – both because it fits the data reasonably well and because the parameters that are important and unimportant in different parts of parameter space make sense. I have two points I would like to make in this review*

We thank the reviewer for her positive feedback and thoughtful comments. We use some of her wording in our revised manuscript.

**First comment** *One is to highlight a potential of DELSA not noted in the paper. In line 117, this paper refers only to the first-order sensitivity capabilities of DELSA. While Rakovec et al (2014) first demonstrated the DELSA approach using first-order sensitivity indices, they also note that the approach has considerable unexplored potential for evaluation of parameter interactions. This requires that the sensitivity matrix include the prior information used for first-order statistics, and also derivatives related to observations, as noted in Rakovec et al (2014, paragraphs 11, 20, 66 and 67, and Figure 12 and Appendix A), and Hill and Tiedeman (2007, Appendix B). The addition of observations in the sensitivity matrix allows calculation of statistics that address concerns such as those considered in Fig. 5 of this work. Commonly this is called a Value of Improved Information (VOII) analysis, and statistics such as OPR (Observation-PRediction) and PPR (Parameter-PRediction) could be used to explore the distribution of uncertainty measures throughout parameter space using the DELSA approach. OPR and PPR are described by Tiedeman et al. (2003, 2004), and Tonkin et al (2007). Parameter-value dependence of these or other statistics with similar goals has received little attention to my knowledge. I imagine that the analyses in this article and those suggested are a small beginning of a future that will see models of complex processes used in ways we*

*early modelers can scarcely imagine. We are riding on horseback while in the future there will be progressively more insightful ways to regard models and integrate their insights into society. This is what I imagine. I am excited that I might live to see what will happen in a few coming decades.*

In response to the reviewer's comment, we plan to make the following changes to the main text:

1- Section 3.2 ( Methods - Sensitivity analysis) we change the second paragraph:

"In this work, we apply the DELSA method (Rakovec et al., 2014), which is a frugal local-global hybrid technique, to identify the parameters that have the largest impact on simulated debris flow variables. Although our implementation only examines first-order sensitivities across the parameter space - as in Rakovec et al. (2014) - it should be noted that DELSA has considerable unexplored potential to characterize parameter interactions, which could be achieved by including additional terms in the prediction total variance, as suggested by Sobol' and Kucherenko (2010)".

2- Section 4.4 (Results and discussion: Parameter identifiability)

The main goal of this study was to characterize the sensitivity of model responses to variations in uncertain rheological parameters, using only independent information on parameter values (i.e., the situation comparable to Sobol', as proposed by Rakovec et al., 2014). Hence, the only verification dataset available (flood area and sediment volume from the March 2015 event) was used to examine the identifiability of model parameters. However, the relative importance of additional observations could be assessed through the Observation-Prediction (OPR) statistic (Tiedeman et al., 2004), and the potential new information provided by field data (e.g. sedimentological and morphological characteristics) for a specific parameter (e.g. alpha1, beta1) could be quantified with the Parameter-Prediction (PPR) statistic (Tonkin et al., 2007). It should be noted that, in both cases, the equation for the total local variance (Equation 7) would be different as additional information should be incorporated (see Appendix A in

Rakovec et al. 2014).

3- Section 5 ( Summary and conclusions) we change the last paragraph:

"Future investigations should advocate for improving the structure of debris flow models to achieve better simulations of deposition/erosion processes, stopping phases, and changing rheologies. Further, the development of computationally frugal methods to improve understanding of parameter interactions in environmental models emerges as an attractive avenue for future research."

**Second comment**

*My second comment is much shorter. In line 215 of the article we find the text "Although sediment concentration is one of the main parameters controlling debris flow rheology, model results are insensitive to Cvmax. This is explained by its small range of variation compared to the feasible range of beta-1, ..." I can see how a narrow parameter range can explain uniformity in parameter sensitivity, but fail to see how it can explain its insensitivity. Perhaps there is something not quite explained well here.*

We thank the reviewer for this observation. Since the original sentence is not a proper explanation of Cvmax sensitivities, we have decided to reword the text as follows:

"The large sensitivities in model response to variations of beta-1 suggest that the viscous stress (second term in equation 3) is the main contributor to sensitivities in simulated frictional slope. On the other hand, DELSA sensitivity indices associated with yield stress ($\tau_y$) and Manning's roughness coefficient - the other terms friction slope -, are of second-order importance. Interestingly, model results are insensitive to Manning's roughness coefficient."

*I appreciate the opportunity to provide comments on this very fine paper. I hope my comments provoke a bit and are of some utility.*

We deeply appreciate the reviewer's thoughtful comments that will help not only to improve our manuscript, but also bring new ideas on potential improvements and applications of sensitivity analysis methods.